



# Brief communication: Landslides on the Argentinian Santa Cruz river mega dam works revealed by PSI DInSAR

Guillermo Tamburini-Beliveau[1], Sebastián Balbarani[2], and Oriol Monserrat[3]

[1] Consejo Nacional de Investigaciones Científicas y Técnicas (CONICET) - Centro de Investigaciones y Transferencia (CIT) de Santa Cruz. Av. Lisandro de la Torre 860, Río Gallegos, Argentina
[2] Departamento de Agrimensura, Facultad de Ingeniería, Universidad de Buenos Aires y Facultad de Ingeniería del Ejército, Universidad de la Defensa Nacional, Ciudad Autónoma de Buenos Aires, Argentina
[3] Geomatics Research Unit, Centre Tecnològic de Telecomunicacions de Catalunya (CTTC/CERCA), Av. Gauss, 7, E-08860 Castelldefels (Barcelona), Spain

**Correspondence:** Guillermo Tamburini-Beliveau (guillermotb@conicet.gob.ar)

**Abstract.** In the dam industry, safety and environmental aspects are crucial beyond production goals. By monitoring landslides associated with the construction of a hydro-power dam in the Santa Cruz River in the Argentine Patagonia, this paper contributes to the assessment of the project safety. Ground deformation is monitored using SAR satellite data and the Persistent Scatterer Interferometry technique, and contrasted with optical imagery, geological and technical reports and fieldwork. The
results include maps of accumulated deformation and deformation time series for the locations of the anchorages of the dam, providing a new and independent dataset to assess the dam work integrity.

## 1 Introduction

Infrastructure and natural hazard monitoring is nowadays a common application of remote sensing. Large civil works and landslides are among some of the most studied elements (Shugar et al., 2021; Moya et al., 2021; Herrera et al., 2009) and,

among the many possible techniques, SAR techniques and, in particular, differential satellite interferometry SAR (DInSAR) play a key role. The significant increase of data availability and the improvements on data processing and analysis make it possible to measure earth surface deformations with precision, reliability and chronological persistence, which have been unheard of until recently. That is, it is therefore possible to assess the earth surface deformation of millimetre magnitudes for periods of several years and with observations every few days through satellite monitoring (Crosetto et al., 2010; Chang, 2015;

Devanthéry et al., 2018).

Regarding the remote sensors monitoring of civil works, dams and their possible structural failure are of particular interest. Despite being considered safe infrastructures, unfortunately and due to various causes, there are many examples of catastrophic failures throughout the planet (Shugar et al., 2021; Lumbroso et al., 2020; Genevois and Ghirotti, 2005; Xu et al., 2008), many of them listed on the web *Lessons Learned from Dam Incidents and Failures*: https://damfailures.org [accessed: August 2022].

Because of the fatal consequences that these failures may have, it is of utmost importance to closely monitor their structural safety.



In this research work, satellite data from the Sentinel-1 and Persistent Scatterer Interferometry (PSI) (Crosetto et al., 2016) are used to map ground displacements associated with the construction of a mega dam in Argentine Patagonia which suffered from serious structural problems during the still on-going construction phase (IEASA, 2020). The work carried out in this study has made it possible to independently monitor such problems, and thus, to assess possible safety risks.

## 2   Dam project location

At latitude 50° South in the Argentine Austral Patagonia (Fig. 1 central planisphere), a large part of the thaw waters of the great Southern Patagonian Ice Field in the Andes and the great Patagonian lakes drain into the Atlantic through the Santa Cruz River. This river forms the backbone of a purely glacial allochthonous basin with global scale dimensions (Grill et al., 2019; Masiokas et al., 2019). It spreads across 30,000 $km^2$ and is in a high state of environmental conservation. The river, located in the geological province of the Austral Basin, flows for 380 km from west to east, forming meanders with a large radius of curvature, and is dominated by an arid landscape of plateau and slightly undulating tertiary sedimentary strata. In the river valley, there are also some occasional ceilings of igneous deposits or Quaternary glaciolacustrine sediments (Astini et al., 2015).

## 3   Hydroelectric Developments of the Santa Cruz River Project

The Hydroelectric Developments of the Santa Cruz River (AHRSC in Spanish) consist of a hydroelectric project for the simultaneous construction of two mega hydroelectric dams on that river. Once completed, its reservoirs will flood 50% of the course of the river. This is the largest infrastructure project in Argentina in recent decades, the largest dam project under construction by Chinese companies outside China, and the southernmost mega dam on the planet, with a minimum cost of 4.5 billion dollars (Tamburini-Beliveau and Folguera, 2022).

Once built, the dams will have a joint installed capacity of 1,310 megawatts with an average annual plant generation of approximately 5,000 GW/h. This will contribute to 2.5% -5% of the maximum peak of national electricity consumption (EBISA, 2017).

Both dams are concrete face rockfill dam (CFRD) type. The Cóndor Cliff - Néstor Kirchner dam (CC-NK), the larger of the two dams and the one studied in this work (Fig. 1) is 68 m high from its crest to the bottom of the channel, and will store 5.3 $km^3$ of water. The second dam, La Barrancosa - Jorge Cepernic, which is downstream from CC-NK, is 41 m high and will generate a 3.5 $km^3$ reservoir (EBISA, 2017).

## 4   Geological framework and problems of the Cóndor Cliff - Néstor Kirchner (CC-NK) mega dam

According to official documentation (IEASA, 2020), at the end of 2018 and beginning of 2019, during the foundation excavations of the main wall of the CC-NK dam, several landslides appeared in the areas of the abutments on both banks of the river, putting the safety of the work at risk. This led to the interruption of the construction works and the complete redesign



and modification of the dam. The new design keeps its productive characteristics, but with a completely different morphology (Fig. 1 A and B). Because of this situation, project deadlines and costs have substantially changed, affecting its financial viability. Large sectors of the work which had already been done until 2019 later had to be removed to readapt the project since

2020 onwards. The structure has been substantially modified : without moving the central sector of the dam axis, its anchors have been diverted a few hundred meters upstream, also readapting the location of other key dam elements (deviation channel, machines hall, etc.). This results in a roughly U-shaped dam very different from the original.

The CC-NK dam is in an area with a geological reality that makes its implementation difficult. It is located in a canyon valley covered by loose Quaternary glaciofluvial sediments from 5 m to more than 60 m thick, and accumulations of material from

multiple mass-wasting events and paleo-slides from the same period (see geomorphological map in Fig. 1). In some elevated sectors - in the tops of the plateaus on the cliffs on the margins of the river canyon - Pliocene basaltic flows also cover the surface (Astini et al., 2015; EBISA, 2017; IEASA, 2020, 2021).

All these materials are found over the Tertiary Santa Cruz formation, which consists of a lower Miocene homocline inclined a few degrees to the SE. It is formed by fluvio-lacustrine sedimentary strata that are very heterogeneous in the vertical direction

(constant and transitional alternation of sandy and pelites packages), which are weak and poorly compacted due to their low diagenesis (Astini et al., 2015; Giambastiani and Filloy, 2018; Tamburini-Beliveau, 2022; Celli and Falcioni, 2022). From the Cenozoic to the last Pleistocene glacial period, this formation has been eroded and fractured by the action of large glaciers, generating glaciotectonic faults and diaclases of various magnitudes (Strelin and C., 1996; Goyanes, 2012).

Additionally, the ceiling of rigid, dense and compact basaltic lava flows, which are present mainly on the northern flank of

the river, favors the occurrence of mass-waste processes due to the strong vertical contrasts between the physical characteristics of the multiple rock strata (Turazzini, 2002; Astini et al., 2015; EBISA, 2017).

Lastly, in the southern sector of the construction site, the rock (Santa Cruz formation) is too far from the surface, covered by glaciofluvial deposits, and consequently, the structure of the dam has serious difficulties for securing the foundations on (the weak) rock. On the north flank, a fault is described that extends a few meters in the upper sector of the location of the axis and

foundation of the dam (Capdevila et al., 2007; IEASA, 2020; Turazzini, 2002).

From the elements presented above, it can be deduced that a difficult and complex geotechnical scenario affects the CC-NK dam and its foundations on both flanks of the wall (Beigt et al., 2016; Genevois and Ghirotti, 2005; Pánek et al., 2018).

## 5   Methods and data

The anchorage area of the abutments of the CC-NK dam, where the slopes instabilities appeared, have been analyzed with

Persistent Scatterer Interferometry (PSI) during the period ranging from November 2018 to April 2021. The particular PSI processing chain and algorithm used in this work is that of the PSIG approach (Devanthéry et al., 2014), developed by the Geomatics Division of the Centre Tecnològic de Telecomunicacions de Catalunya (PSIG-CTTC), and the input source data have been acquired by the Sentinel 1 SAR satellite constellation (see Tab. 1-A and Appendix A).



**Figure 1.** A dot in the very center of the planisphere (between A and B) shows the project location (50.21° South, 70.78° West). A: First design of the CC-NK dam (2015). B: Second design of the CC-NK dam after experiencing structural issues (2020). At the background, the geomorphological map with the following categories may be observed: 1, Alluvial plain. 2, Fluvial terraces. 3, Active alluvial fan. 4, Flank pediment. 5, Basal moraine. 6, Marginal moraine. 7, Glaciofluvial terrace covered by catastrophic flow deposits. 8, Rock avalanche deposits. 9, lateral spread and rotational landslides. 10, Rotational landslides. 11, Scree slope. 12, Rock fall. 13, Lava flow. 14, Seasonal lagoon. 15, Wind blowout. 16, Parabolic dune. 17, Santa Cruz river channel. Box C shows the optical satellite image where the cracks of the south bank can be seen. In D, a view from the ground of the crack on the south margin may be observed. Source: Our own elaboration based on (EBISA, 2017; IEASA, 2020), © Google Maps, www.worldmapgenerator.com [accessed August 2022] and on fieldwork.



The satellite DinSAR method consists of the exploitation of the observation of the phase differences measured in the radar
signal between two satellite SAR images of the same area acquired at different times and from slightly different positions. The
measured interferometric phase represents differences along the satellite-target travel of the radar signal. After the necessary
corrections and parameterizations, the movement of the ground target between both acquisitions may be estimated.

The PSI method is an advanced DInSAR technique that exploits stacks of images acquired at different times of the same
area to provide high precision ground (or ground fixed infrastructures) displacement measurements. This technique allows
robust measurements along time providing displacement time series and velocity estimations for the location of each one of
the measured points. By "points" we refer to pixels of a satellite raster image representing a small portion of the surface
of the Earth. The final PSI time series contains the accumulated magnitude of the deformation in correspondence with the
time-ordered SAR images starting from a reference image. For more details about PSI techniques, see Crosetto et al. (2016).

Optical images have been used to monitor the evolution of the works and double check what was revealed by the PSI results
(see Tab. 1-B). Other digital vector mapping data (geological and geomorphological maps, contour lines and plans of the dams)
have been obtained from the dam's environmental impact assessment report (EBISA, 2017), other official technical documents
(IEASA, 2020, 2021; Astini et al., 2015) and an online cartography server: https://observatorio.ieasa.com.ar/geovisor.php [Accessed: August 2022]. The information has been collected and analyzed with the QGIS software (Quantum GIS Development
Team, 2022).

## 6   Results

The main result of this work consists of a map of accumulated ground displacement in the CC-NK dam area and surroundings
(Fig. 2 and Appendix B1 and B2) and the displacement time series from November 2018 to April 2021. All the presented
measurements are in Line of Sight (LOS).

A total of 68477 Persistent Scatterers (PSs) were obtained in a region of 37.1 km$^2$. This corresponds to a ratio of 1897
PSs/km$^2$, approximately one point every 527m$^2$ (see Figs. 2-X and B2). Out of these, 1018 PSs (0,7%) show a displacement
above $3\sigma$ ($\sigma$=2.69), i.e. an accumulated deformation over 8.07 mm along the monitored period. The value of $3\sigma$ was taken to
clearly discriminate those points with random scattered values around 0 from those with clear deformation trends (see Fig. B1-E). Consequently, to better focus the analysis, the first group of $<3\sigma$ was discarded. Thus, only the reflectors with pronounced
deformation trends have been analysed (Fig. 2-X).

From this initial dataset, the sub region that includes the work of the CC-NK dam was analysed in detail (Fig. 2-Y; small
polygon on Fig. 2-X). This corresponds approximately to 9.6 km$^2$. There, 15678 PSs were obtained, that is 1742 PSs/km$^2$,
approximately one point every 574 m$^2$, out of which 857 (5.5%) have magnitudes of $>3\sigma$ of the original dataset.

The 5.5% of $>3\sigma$ PSs of the detailed area of interest versus the 0.7% of the whole region reveals the evident concentration
of ground displacement phenomena in the dam work area.

The results allow us to identify five types of characteristic data:

1. Zones in the range of negligible displacements ($<3\sigma$ already mentioned).


**Figure 2.** Top left (X). Area covered by the PSI processing. White polygons delimitate the two mentioned areas, the general starting dataset and the in detail analysed zone. The optical background image is from Google from July 2022 (© Google Maps). Coordinates are UTM zone 19S. Only reflectors >3$\sigma$ are represented, magenta negative and cian positive deviation, scale ranges until 13$\sigma$. Top right (Y): construction site. The plan of the second dam design has been superimposed. The PSs of each landslide subgroup are represented on the southern (A, B) and northern (C, D) margins. Charts A to D correspond with the deformation time series of the calculated average for each one of the landslide areas (A to D).





| Remote sensors sources | |
| --- | --- |
| **A) Characteristics of the SAR images used in the PSI process** | |
| Platform | Sentinel-1B |
| Sensor | C-SAR |
| Acquisition mode | Interferometric Wide (IW) Swath |
| Temporal span | 29 October 2018 - 28 April 2021 (2.4 years) |
| Number of images | 77 |
| Wavelength | 0.055 m |
| Acquisition mode | Interferometric Wide (IW) Swath |
| Product type | Single Looked Complex (SLC) |
| Orbit | Descending |
| Incidence angle | 32° |
| Track or relative orbit number | path 39 orbit 25790 |
| Minimum revisit period | 12 days |
| Polarization | VV |
| Full resolution (azimuth/range) | 5 m x 20 m |
| Number of processed swaths | 1 (Swath 02) |
| Number of processed bursts | 1 (Burst 08) |
| **B) Basic characteristics of the optical sensors true color band combination** | |
| 92 Planet images | 3 m resolution |
| 3 Sentinel images | 10.5 m resolution |
| 2 Google Image | 0.5 m / 0.3 m |

**Table 1.** Main characteristics of the input satellite data for the research (Our own elaboration).

2. Landslides on slopes, the ones under analysis in this work (see Fig. 2).

3. Foreseeable deformation trends associated with the works. For example, soil uplift due to the progressive construction of a retaining slope, or subsidence due to the extraction of aggregates.

4. Systematic or random errors. Mainly phase jump or atmospheric errors, respectively.

5. Absence of measurements, i.e. PSs, due to modification of the terrain due to the works. In the face of substantial changes in the surface, the possibility of having persistent reflectors is lost.

This paper focuses on item 2 of this list. More details on the characteristics of the other items may be found in the Appendix B.





## 7  Discussion


The official reports (IEASA, 2020, 2021) indicate that the two areas of the dam abutments, on both the north and south banks of the river, are affected by landslides. More precisely, PSI measurements made it possible to identify two subareas with their own deformation dynamics on each riverbank. All the PSs within each subarea were averaged, obtaining a single PSI value per subarea. The analysis of the optical images confirmed that these areas were not affected by significant changes in the ground

surface during the monitored period. The largest of these areas approximately occupies an area of 2 ha., and the smallest, of 0.2 ha.

The results are presented in Fig. 2. On the south bank, two movement zones are identified, zone A with 51 (red) reflector points and zone B with 18 (blue). On the north bank the identified zones are: zone C with 33 reflecting points (yellow) and D with 15 (orange).

Maximum accumulated displacements between extreme values range from 40 to 80 mm. This represents 16 mm/yr for area C, and 32 mm/yr in area D. A polynomial fit line with a determination coefficient $R^2$ (Mirmazloumi et al., 2022) has been superimposed in the chart to show the absence of noise in the measurements. It is worth underlining that the magnitudes measured do not accurately reflect the real magnitude of the movements since:

1. PSI provides measurements along the LOS component, which only reflects part (a projection) of the real displacement
vector,

2. PSI is opportunistic, that is, it only provides measurements on those points that provide permanent response to the satellite,

3. The period analysed does not cover the entire displacement period.

Area A is located in a flat area, so it could be said that most of the movement is vertical, which, according to the orbit

direction of the satellite, is in agreement with the negative values of the LOS measurement showing subsidence.

Area B is located on the slope, so horizontal and vertical displacements are measured. The general orientation of this area is not the optimal to be observed with the descending satellite trajectory, but the considerable magnitude of the deformation made it possible to obtain results. Presumably, deformation to the west and east of the A-B locations became invisible to the radar either by the large magnitude of the ground displacements (at west, in the crack, where data decorrelation is present, see

Fig. B1-I in Appendix B) or due to the deficient orientation of the slope and the smaller magnitude of the displacements (to the east).

Observing the situation on sectors A and B (with inverse trend sign), it is possible to suggest that a rotational mass wasting is taking place, which agrees with the bibliography (Astini et al., 2015), but more information is necessary to provide a more definitive conclusion.

Since area C is mainly flat, it is possible to say that a vertical component in the ground deformation is relevant, but the PSI information is not enough to determine the exact direction of the deformation vector. In the chart, a step in the trend is visible in October 2019, which coincides with the start of containment works in the area as revealed by the optical sources.



Finally, as sector D is found in a steep surface, both vertical and horizontal displacements are relevant, but further measurements should be made to get a complete comprehension of the situation there. The strong slope of the curve around February
2019 confirms the ground instability trends described in the company reports (IEASA, 2020).

In none of the four curves can the arrest of surface deformations be clearly seen, so it is possible to affirm that the sliding processes were still active in April 2021 (date of the last observation). The works in general, and the movements in particular, are clearly located in areas of Quaternary sedimentation and materials resulting from paleo mass wasting processes that had already been described as potentially conflictive in the preliminary project reports (see Fig. 1 A and B), such as:

– on the north bank: basal moraine (8), lateral expansion and rotational slide (18), rotational slide (20);

– and on the south bank: marginal moraine (9) and lateral expansion and rotational slide (18).

The ground dynamics described in this piece of research coincide with what may be consulted in the reports from the company contracting the work (IEASA, 2020), in which the landslides are associated with the reactivation of a paleo-slide in the southern sector, and to the presence of a subhorizontal stratum whose material presented very low resistance and high
plasticity in the northern sector.

## 8   Conclusions

Through the use of the DinSAR PSI technique, it has been possible to confirm that, as indicated by civil organizations, the media and, finally, the official reports, slope instability processes have been activated or reactivated by the construction works, putting the integrity of the dam wall anchorages at risk. This has been corroborated with optical satellite imagery and field
work. According to official information, the period studied should cover the beginning and the end of the slides. However, the trend charts do not confirm that the movements have stopped.

We have presented relevant unpublished information regarding the integrity and safety of a large-scale civil work of international importance, since the Chinese and the Argentine governments and a world-class hydrographic basin are involved. Moreover, it is relevant to note that the monitoring has been performed over an ongoing project, allowing its tracking while it
is being built, and not over a finished and already operative work, as can usually be found in the bibliography.

It is necessary to anticipate this type of problems both because of the economic costs and because of the safety problems that they can lead to. With this work, we have demonstrated that it is possible to effectively and rapidly contribute to this subject by applying remote sensing techniques. Thus, these techniques offer useful information for management authorities and local communities to independently assess the current state of projects of this kind.

In this particular case, and with a minimum investment cost, we have been able to independently and remotely monitor a serious eventuality with potential dramatic consequences on a mega dam project located in a remote region. The weaknesses in the anchorages on both sides of a dam wall of an hydropower facility system with a capacity for 8.8 km$^3$ of water could be catastrophic if not detected in time.



Finally, and attending the magnitude of the experienced geotechnical problems, we hope we have made a valuable contribu-
tion to the adequate and safe evolution of the AHRSC project, which still remains situated in a very challenging terrain.

## Appendix A: Methods

We implemented the advanced differential interferometry SAR algorithm named Persistent Scatterer Interferometry (PSI)
which is particularly suitable for monitoring outcrops, urban areas and man-made infrastructures (Devanthéry et al., 2014).
The processing steps include: 1) Data Extraction. 2) DEM Preparation. 3) Initial Processing. 4) Processing of velocity. 5)
Phase unwrapping. 6) Geocoding.

A temporal baseline limit of 100 days has been applied to the interferometric pairs. Due to the small orbital tunnel config-
uration of the Sentinel-1 mission (i. e. short spatial baseline), the most important decorrelation factor comes from the surface
changes along time. No perpendicular baseline limit has been applied.

77 high-resolution SAR scenes have been used. The time spanned from 29 October 2018 to 28 April 2021 (2.4 years), with
a 12-day revisit time. Single Look Complex (SLC) product and Interferometric Wide (IW) acquisition mode were selected and
downloaded from the on-line ESA Copernicus Open Access Hub. The precise orbit files were downloaded from the Copernicus
Sentinels POD Data Hub.

For each SAR image, the co-polar vertical-vertical (VV) channel was extracted from the original data in double polarization
(VV+VH) and the corresponding swath of the area of interest (IW2). The 30 meters spatial resolution digital elevation model
of Shuttle Radar Topography Mission (SRTM) was used to calculate the topographic component that was later extracted. The
DEM tiles were downloaded from the United States Geological Survey (USGS) EarthExplorer catalog.

A simulation of the digital elevation model for the SAR geometry of the super-master image was carried out. So, interfero-
metric pairs and corregistration were created between them. Interferometric phase quality indicators were created, such as the
amplitude and the average dispersion. We generated 580 high-resolution differential interferograms. The deformation velocity
and the topographic error (or residual) were estimated. Then, the selection of the suitable points where the final solution would
be obtained was carried out. Other parameters defined for the PSs inclusion for the final results dataset were: 1) a maximum
threshold of the average dispersion of the amplitude of 0.50 and, 2) a temporal coherence maximum value of 0.65.

Finally, the interferometric phase unwrapping was performed, obtaining the absolute deformation values for each PSs. The
final coordinates of the PSs are offered in a global geographic coordinate system, based on a reference ellipsoid. Thereby,
a georeferenced point cloud was obtained with information on the absolute displacements for each acquisition date and the
deformation velocity, which allowed us to display, compare and analyze the deformation time series in a geographic information
system (GIS).





## Appendix B: Supplemental Figures

Figures B1 and B2 show examples of PSs with high $\sigma$ values that have not been included in this work and correspond to items

mentioned in page 5 and following briefly described. The letter notation is the same for both figures:

- E. Frequency distribution graph of the accumulated displacement of the PSs reflectors in the study area and the $3\sigma$ threshold.

- F. Time series of a stable PS with dispersed measurements due to random errors. Elevated $\sigma$ between 2 and 3 due, $R^2$=0.082.

- G. Phase unwrapping error (PUE). 40 reflectors average. Red dots in the chart in Fig. B1 represent the series after applying a manual PUE correction jump of 28 mm (Mirmazloumi et al., 2022).

- H. Deformation trend (Fig. B1-H1) caused by construction works (B1-H2). 72 reflectors average.

- I. 11 phase unwrapping errors for a single reflector located in the root of the big crack in the southern margin (west of studied sectors A and B). This is an exceptional and isolated sample which preserved the coherence along the time

series despite the high deformation rate. Chart in Fig. B1-I2 indicates the deformation trend without PUE correction. I3 corresponds to the same chart after applying a manual PUE correction. A visual detail of the crack is shown in Fig. B1-I.3 and the location of that reflector in Fig. B2 as a grey diamond. Official documentation describes the occurrence of the landslide on 14th February 2019. This is clearly visible in the charts (first vertical dashed red line). At the same time, it is clear that the trend was present before (at lower rates) and after this date. Dashed red vertical lines indicate the en-

forcement of a PUE correction, eleven times in this case. Official reports (IEASA, 2021) describe ground displacements up to 15 mm/day (5.5 m/yr) in the main sector of this big crack, where we could not obtain PSI. Observing this chart, we cannot confirm the end of the deformation trend in April 2021.

Fig. B3 presents three stages of the project evolution through optical sensors where main project changes can be seen.
**Figure B1.** Examples of PSs with high $\sigma$ values which, as presented in page 5 and following the same number notation, have not been studied in this work. Optical satellite images from © Google Maps

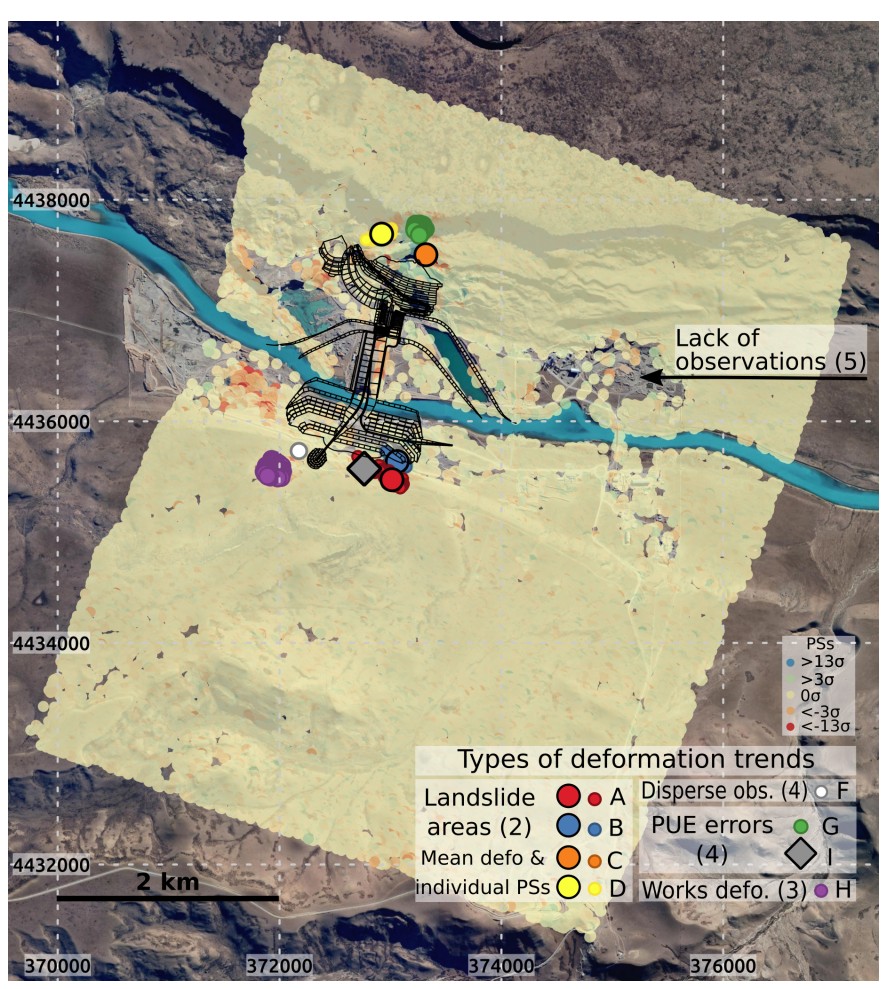

**Figure B2.** Map of the full PSI results and examples of the identified deformation trends, 1 to 5, as described on the items on page 5 or presented in previous B1. A to D are landslides studied in this work. F to I are the other time series trends briefly described in this Appendix. Arrow (number 5 as in page 5) is an example of the absence of observations due to the surface changes produced by the construction work. Our own elaboration. Background satellite image © Google Maps

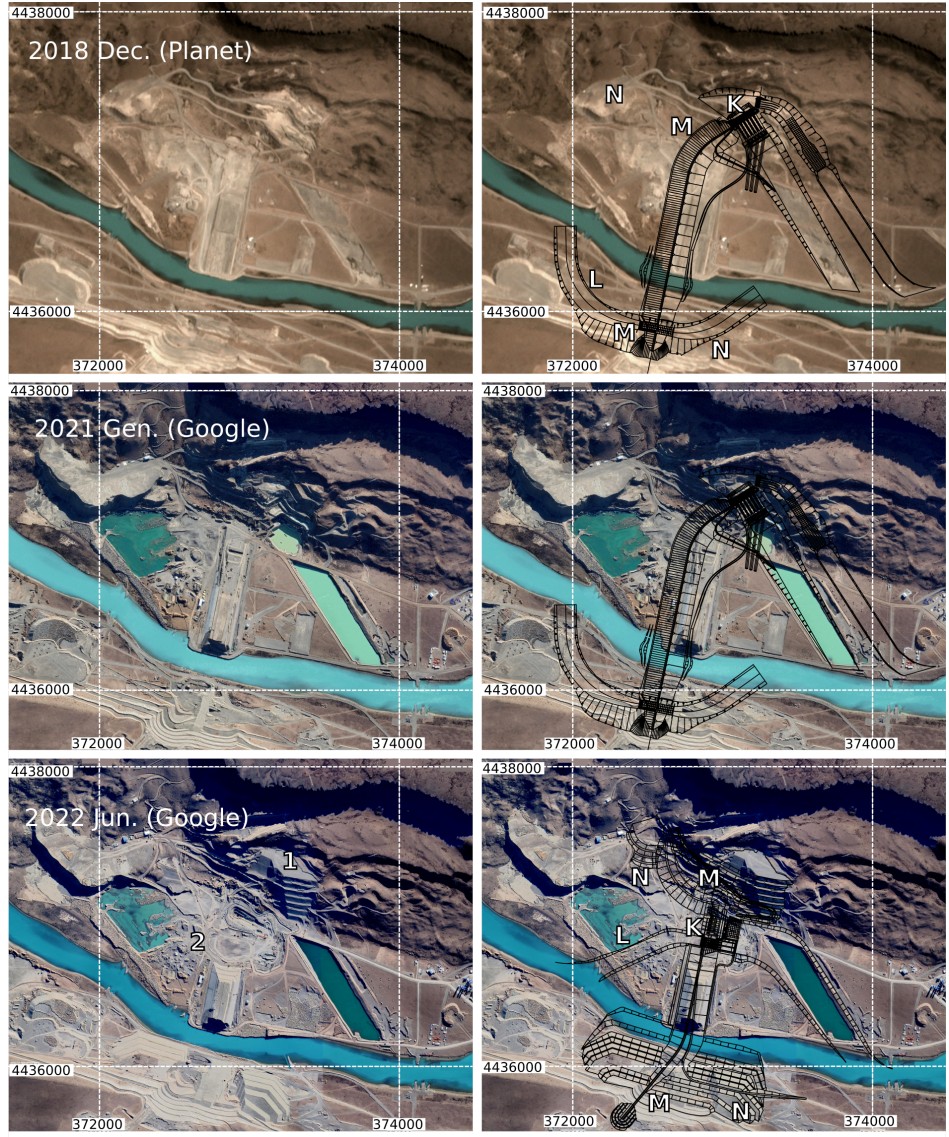

**Figure B3.** Three moments of the dam construction. In the left column, optical image, in the right, the dam plans have been superimposed (the third row corresponds to the new project). Letters identify main structural changes: K) machines hall, L) deviation channel, M) change of the dam wall direction, N) structural retaining walls. Numbers identify main correction works to the original project (modification of structures already built): 1) the space prepared to the main hall have been filled in again (over 1 million m$^3$), 2) an important part of the dam wall (which was originally planned to have 12.1 millions m$^3$) has been retired, and a big 10 m wide and 36 m deep exploratory well is built there to study the rock substratum. To dig the well, a volume of around 70.000 m$^3$ had to be previously removed at the ground level in the area which had already been filled with basement materials of the dam wall. Our own elaboration. Background images from Planet$^{TM}$ and ©
Google Maps.



*Author contributions.* G. T. B. coordinated the team, wrote the text, designed the figures and performed the general research over the study

case: geological setting, engineering and environmental impact assessment, technical reports analysis, available data, etc. S. B. processed the

PSI data and developed the methods section. O. M. carried out a general supervision of the research work.

*Competing interests.* The authors declare that they have no conflict of interest.

*Acknowledgements.* Thanks to LaIC (Laboratorio de Inglés Científico - UNPA) for their linguistic support and corrections of this manuscript.



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
