# Peer review of "Brief communication: Landslides on the Argentinian Santa Cruz river mega dam works revealed by PSI DInSAR"

_Natural Hazards and Earth System Sciences, 2022_

## Author Response (AR1)

**Brief communication: Landslide activity on the Argentinian Santa Cruz river mega dam works confirmed by PSI DInSAR - Author's response**

14th April 2023

Dear Editor,

As the text has been edited in an online platform in LaTex, we could not apply changes tracking. However, next you will find the precise answer to the referee's requests with clear indications to its corrections pointing to the line number in the final PDF text version.

In the following explanation, the answer to the referees is written in italics font after the ":" after the original referee comment.

Minor editing corrections required in the very first submission steps, such as order in the number labels in figures, some random words appearing in the text (due to the LaTex formatting), copyright symbol, and others, have already been solved. On the other hand, a final professional language correction has been applied.

Finally, the corresponding author's email has been changed to: guitambe@fceia.unr.edu.ar

**FIRST REFEREE**

General comments:

- Title, terms "confirm" and "activity": *applied*.
- More about the contributions on assessing the project safety: *Some thoughts have been added on this subject in the Conclusions section. Lines 189 to 197 in the new text version*.
- It could be useful to highlight the work phases on panels A-D of figure 2 so that a comparison between the trend and the work's process can be done: *Figure 2 has been modified and now a timeline is present to better show the processes*.

Particular corrections:

- Ln41 (now 39), MW instead of megawatt: *Done*.
- Ln74, is the fault within the image boundaries? If yes, please add it in the figure. If no, please specify it in the text: *The fault is within the boundaries, but the precise location is not available. A comment has been added in that sense in the text*.
- Figure 1, numbers on purple and dark green are not so visible. Moreover, as in Figure 2, I'll suggest to use "Coordinates are UTM zone 19S" instead of "50.21° South, 70.78° West": *changes applied*.
- Ln104 (now 106), use 68,477 to be consistent with 5,000 at ln42: *Consistency applied to this case and others in the whole text*.
- Table 1, caption should be placed above the table and not below: *We used the Latex journal template. We left that minor trouble for the editing services to avoid discordances between LaTex commands*.

- References: according to the guideline references allowed for a brief communication are just 20: *Applied*.

**SECOND REFEREE**

Major points:

- Section 5 will benefit from a more rigorous description of the processing approach. I understand that there is not much space to do this in a brief communication, but some general sentences (e.g.lines 87 to 92, now 87 to 95) shall be substituted with information about the processing solution adopted here: *In effect attending to the brief communication format it is no possible to expand about it in the text, and the cited bibliography is very accurate regarding this point. However, some details have been added in this regard*.

Minor points:

- Some acronyms are not defined in the text. Please check: *Ln10, Corrected*.
- I suggest that you use the velocity of PSs along with the accumulated displacement. Velocity is a more readable parameter for non-experts: *The velocity is already mentioned along the text. It has been added in one occasion where it was missing (line 107)*.
- Abstract, first line. The term "dam industry" looks odd to me: *Changed by "hydropower industry" using the following reference*:
  https://guides.loc.gov/renewable-energy/hydropower
- Line 19. Is it needed to write the title of the website page? *It has been removed*.
- Line 63 (now 59-60), crop out instead of cover the surface: *As lava flows, the mentioned rocks cover the surface instead of cropping out on it*.
- Fig.1. I propose to change the color of the sketch of the dam project from black to white: *Changed*.
- Line 104 (now 106) density instead of ratio: *Now line 106*.
- Line 109 (now 111) 'pronounced deformation trend' is a vague word choice: *Changed by a synonym*.